# Advanced Sign Language Video Generation with Compressed and Quantized Multi-Condition Tokenization

**Cong Wang**[1*]   **Zexuan Deng**[1*]   **Zhiwei Jiang**[1†]   **Yafeng Yin**[1†]   **Fei Shen**[2]
**Zifeng Cheng**[1]   **Shiping Ge**[1]   **Shiwei Gan**[1]   **Qing Gu**[1]
[1] State Key Laboratory for Novel Software Technology, Nanjing University
[2] National University of Singapore
{cw,dengzx}@smail.nju.edu.cn   {jzw,yafeng}@nju.edu.cn
shenfei29@nus.edu.sg   {chengzf,shipingge,sw}@smail.nju.edu.cn
guq@nju.edu.cn

## Abstract

Sign Language Video Generation (SLVG) seeks to generate identity-preserving sign language videos from spoken language texts. Existing methods primarily rely on the single coarse condition (*e.g.*, skeleton sequences) as the intermediary to bridge the translation model and the video generation model, which limits both the naturalness and expressiveness of the generated videos. To overcome these limitations, we propose SignViP, a novel SLVG framework that incorporates multiple fine-grained conditions for improved generation fidelity. Rather than directly translating error-prone high-dimensional conditions, SignViP adopts a discrete tokenization paradigm to integrate and represent fine-grained conditions (*i.e.*, fine-grained poses and 3D hands). SignViP contains three core components. (1) Sign Video Diffusion Model is jointly trained with a multi-condition encoder to learn continuous embeddings that encapsulate fine-grained motion and appearance. (2) Finite Scalar Quantization (FSQ) Autoencoder is further trained to compress and quantize these embeddings into discrete tokens for compact representation of the conditions. (3) Multi-Condition Token Translator is trained to translate spoken language text to discrete multi-condition tokens. During inference, Multi-Condition Token Translator first translates the spoken language text into discrete multi-condition tokens. These tokens are then decoded to continuous embeddings by FSQ Autoencoder, which are subsequently injected into Sign Video Diffusion Model to guide video generation. Experimental results show that SignViP achieves state-of-the-art performance across metrics, including video quality, temporal coherence, and semantic fidelity. The code is available at https://github.com/umnooob/signvip/.

## 1   Introduction

Sign language, as a visual language, serves as the primary communication medium for deaf individuals. Early research focused on Sign Language Recognition (SLR) [32, 45, 90], Translation (SLT) [5, 20, 42, 14], or Production (SLP) [58, 56, 60, 86, 83]. More recently, Sign Language Video Generation (SLVG) [57, 61, 47, 69] has gained increasing attention, which aims to generate realistic and expressive sign language videos from spoken language texts, preserving the unique identity of a target signer, as

---

*Equal contribution.
†Corresponding author.

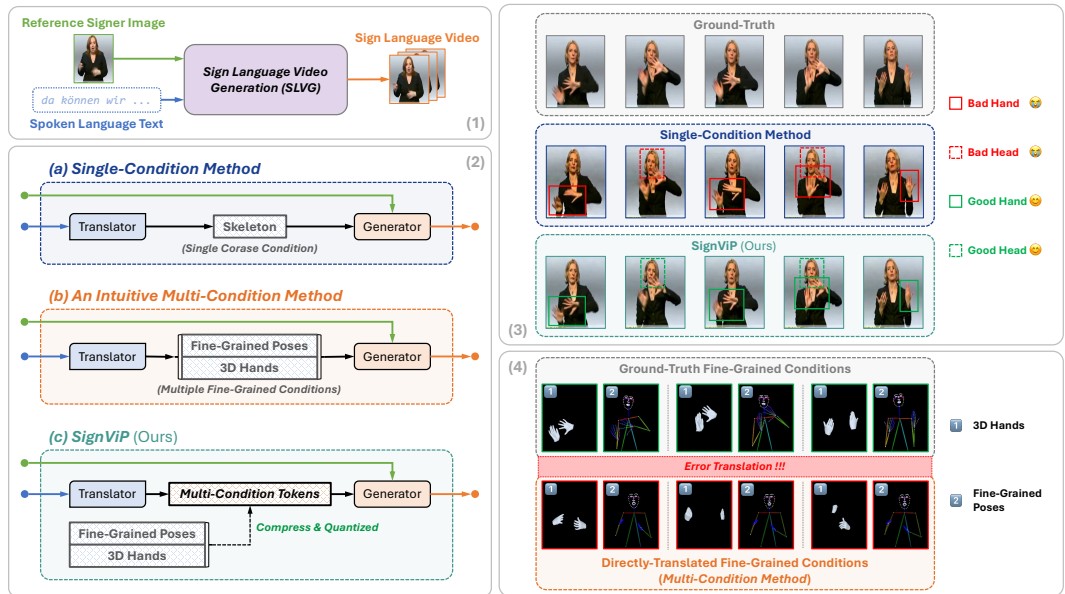

Figure 1: (1) The illustration of SLVG task. (2) The pipeline comparison between existing SLVG methods (*i.e.*, single-condition method and multi-condition method) and our SignViP. (3) Single-condition methods struggle to accurately capture the naturalness and expressiveness of sign language videos. (4) Multi-condition methods are prone to translation errors for fine-grained conditions.

shown in Figure 1(1). This growing interest is driven by its potential applications in accessibility technologies, educational tools, immersive communication systems, *etc.*

SLVG presents a challenging problem due to the lack of explicit spatial or temporal alignment between the input (*i.e.*, the spoken language texts) and output (*i.e.*, the sign language videos) modalities. To address this, current SVLG methods focus on leveraging synchronized auxiliary conditions as an intermediary to align these two modalities. As shown in Figure 1(2a), most of the existing SLVG methods [57, 61] leverage skeletal sequences as an intermediary to bridge a text-to-skeleton translation model (*i.e.*, an SLP model) and a skeleton-to-video generation model. Because the generative model is only guided by a single coarse condition (*e.g.*, skeleton), such single-conditional methods struggle with the naturalness and expressiveness of the generated videos, particularly in capturing facial expressions and figure movements, as illustrated in Figure 1(3).

Recent advancements in human video generation have shown that incorporating fine-grained conditions (*e.g.*, dense pose [84] or 3D models [9, 82]) or leveraging multiple conditions (*e.g.*, depth combined with optical flow [85]) can substantially improve generative fidelity. Inspired by these, we consider whether multiple fine-grained conditions can be introduced to enhance the quality and expressiveness of generated sign language videos. As shown in Figure 1(2b), one intuitive solution is to extend the single-conditional methods by considering multiple fine-grained conditions as intermediaries, where a multi-condition translation model can directly predict capable of achieving multiple fine-grained conditions. However, as shown in Figure 1(4), we observe that directly translating such attributes is challenging due to their high-dimensional nature and susceptibility to errors. This raises an important question: *How can we overcome the challenges in the translation of multiple fine-grained conditions to further advance SLVG?*

To address the challenges, we propose **SignViP**, a novel framework designed to advance SLVG by incorporating multiple fine-grained conditions for enhanced generation fidelity. As shown in Figure 1(2c), instead of directly translating high-dimensional conditions from spoken language texts, SignViP adopts a discrete tokenization paradigm to effectively integrate and represent these fine-grained conditions. Central to this framework is the construction of a **discrete multi-condition token space**, which bridges fine-grained conditions (*e.g.*, fine-grained poses and 3D hands) with the dynamics of sign language video frames. The framework consists of three key components: (1) **Sign Video Diffusion Model** is jointly trained with a multi-condition encoder using denoising loss to

generate continuous embeddings that encapsulate fine-grained motion and appearance details; (2) **Finite Scalar Quantization (FSQ) Autoencoder** is trained with reconstruction loss to compress and quantize the continuous embeddings into discrete tokens, enabling highly dense representation for the conditions; (3) **Multi-Condition Token Translator** is built upon an autoregressive model to translate spoken language text to discrete multi-condition tokens. During inference, the spoken language text is first translated into discrete multi-condition tokens by Multi-Condition Token Translator. These tokens are then decoded back into continuous embeddings by FSQ Autoencoder, which are subsequently injected into Sign Video Diffusion Model to guide sign language video generation. Our experimental results demonstrate that SignViP achieves state-of-the-art performance across multiple evaluation metrics, including video quality, temporal coherence, and semantic fidelity.

Our main contributions are summarized as follows:

- We introduce SignViP, a novel framework for Sign Language Video Generation (SLVG) that incorporates multiple fine-grained conditions for improved video quality and expressiveness.

- We propose a discrete tokenization paradigm through the construction of a discrete multi-condition token space to bridges fine-grained conditions with the dynamics of sign language video frames.

- The experiments validate the effectiveness of SignViP, demonstrating state-of-the-art performance across diverse metrics.

## 2   Related Works

**Sign Language Video Generation.** Sign Language Video Generation (SLVG) aims to generate identity-preserving sign language videos from spoken language texts. Early methods decompose the task into two consecutive sub-tasks [57, 61], which are the text-to-skeleton translation (*i.e.*, SLP) and the skeleton-to-video generation. SignGAN [57] first employs a transformer [76] with a mixture density formulation to translate spoken language text to skeletal sequence. Then, a GAN-based [17] skeleton-conditioned human synthesis model is introduced to generate sign language videos. FS-Net [61] extends SignGAN by predicting the temporal alignment to a continuous signing sequence. Because the single condition focuses solely on capturing basic pose structures while neglecting fine-grained details, the generated videos tend to appear less natural and expressive. SignGen [47] seeks to address this limitation through a novel end-to-end pipeline that integrates multi-condition guidance, including optical flow, pose, and depth. However, SignGen suffers from training-inference inconsistency, which leads to suboptimal results. In this paper, we aim to develop a framework that leverages multiple fine-grained conditions to enhance the quality and expressiveness of generated sign language videos.

**Human Video Generation.** Human video generation has advanced significantly with the deep generative models. Early approaches, such as Pix2PixHD [80] and vid2vid [79], leveraged GANs [17] to generate realistic images and videos from the structured inputs. Several works have also explored the human pose generation, conditioning on the whole body [2, 38, 40, 63], face [10, 33], or hands [73, 35]. However, GAN-based methods often suffer from mode collapse and optimization challenges. More recently, diffusion models [70, 28, 67, 78, 64, 68] have emerged as a robust alternative, producing high-quality images or videos with greater stability. Most prior diffusion-based approaches rely on ControlNet [88] and OpenPose [7] to process each video frame independently, neglecting the temporal consistency and leading to the inevitable flickering artifacts. Pose-guided diffusion models [62, 29, 77, 91, 65, 66] addresses this issue by generating temporally consistent human videos while preserving appearance fidelity. Furthermore, recent research shows that incorporating fine-grained conditions, such as dense pose [84] or 3D models [9, 82], or leveraging multiple complementary conditions, such as depth and optical flow [85], can significantly enhance generative fidelity. Building on these advancements, we aim to harness state-of-the-art diffusion-based methods alongside multiple fine-grained conditions to advance SLVG further.

## 3   Methodology

### 3.1   Preliminary

**Diffusion Models.** As a class of the generative models, the diffusion models [70, 28] consists of two processes, which are the diffusion process and the denoising process, respectively. In the diffusion

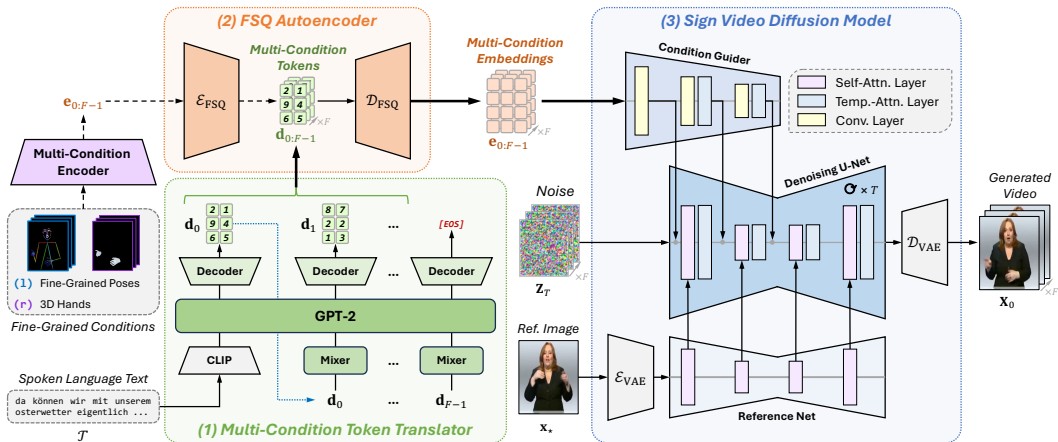

Figure 2: Framework of our SignViP for sign language video generation (SLVG). (1) The spoken language text is translated into the multi-condition tokens by Multi-Condition Token Translator. (2) These tokens are decoded by FSQ Autoencoder into multi-condition embeddings, which are equivalent to the embeddings of multiple fine-grained conditions (*i.e.*, fine-grained poses and 3D hands) generated by a multi-condition encoder. (3) The embeddings are injected into Sign Video Diffusion Model to guide the generation of sign language videos.

process, the Gaussian noise is iteratively added to degrade the input sample over $T$ steps until the sample becomes completely random noise. In the denoising process, a denoising model is used to iteratively generate a sample from the sampled Gaussian noise. When training, given an input sample $\mathbf{x}_0$ and condition $\mathbf{c}$, the denoising loss is defined as

$$\mathcal{L}_{\text{denoise}} = \mathbb{E}_{\mathbf{x}_0, \boldsymbol{\epsilon} \sim \mathcal{N}(\mathbf{0}, \mathbf{I}), \mathbf{c}, t} \left\| \boldsymbol{\epsilon}_\theta \left( \mathbf{x}_t, \mathbf{c}, t \right) - \boldsymbol{\epsilon} \right\|^2 . \tag{1}$$

Among them, $\mathbf{x}_t = \sqrt{\alpha_t} \mathbf{x}_0 + \sqrt{1 - \alpha_t} \boldsymbol{\epsilon}$ is the noisy sample at timestep $t \in [1, T]$, where $\alpha_t$ is a predefined scalar from the noise scheduler. $\boldsymbol{\epsilon}$ is the added noise. $\boldsymbol{\epsilon}_\theta$ is the denoiser with the learnable parameters $\theta$, which predicts the noise to be removed from the noisy sample. Latent Diffusion Models (LDMs) [55] stands out as one of the most popular diffusion models. It performs the two processes in the latent space, which is encoded by a Variational Auto-Encoder (VAE) [31, 54].

**Finite Scalar Quantization.** Finite Scalar Quantization (FSQ) [41] is a concise quantization technique to compress continuous values into discrete values. FSQ can be an alternative of Vector Quantization (VQ) [18] in VQ-VAE [75]. Compared with VQ, FSQ does not suffer from codebook collapse and does not need complex machinery to learn expressive discrete representations. Specifically, given a scalar $z \in \mathbb{R}$ from the encoded latent, the quantized discrete value by FSQ is

$$f(z; L) = (L - 1)\sigma(z) ; \text{ FSQ}(z) \triangleq \text{rnd}(f(z; L)) \in [0, 1, \cdots, L - 1] . \tag{2}$$

Among them, rnd($\cdot$) is round function. $\sigma(\cdot)$ is sigmoid function. $L$ is the predefined quantization level. Through this process, each value $z$ is enumerated, leading to a bijection from $z$ to an integer in $\{0, 1, \ldots, L - 1\}$. For a $d$-dimensional latent vector, the total codebook size is the product of $L_i$ across all dimensions, resulting in $\prod_{i=1}^{d} L_i$ possible discrete representations.

## 3.2 Overview

Given a reference signer image $\mathbf{x}_\star$ and a spoken language text $\mathcal{T}$, SLVG aims to generate a sign language video $\mathbf{X}$ with $F$ frames, where the signer performs sign language accurately aligned with the semantics of the spoken language text. Specifically, the SLVG task can be formulated as $p_\theta(\mathbf{X}|\mathbf{x}_\star, \mathcal{T})$, where $\theta$ is parameters of the SLVG model.

Due to the lack of explicit spatial or temporal alignment between $\mathcal{T}$ and $\mathbf{X}$, current SVLG methods focus on leveraging synchronized auxiliary conditions (*e.g.*, skeletal sequence) as an intermediary to align them. Such pipeline paradigm can be formulated as

$$p_\theta(\mathbf{X}|\mathbf{x}_\star, \mathcal{T}) = p_{\theta_{\text{gen}}}(\mathbf{X}|\mathbf{x}_\star, \mathbf{C}) \cdot p_{\theta_{\text{tran}}}(\mathbf{C}|\mathcal{T}) . \tag{3}$$

Among them, $\mathbf{C}$ is the synchronized auxiliary conditions which spatially and temporally aligned with the target video frames. $\theta_{\text{tran}}$ is parameters of the text-to-condition translation model, while $\theta_{\text{gen}}$ is parameters of condition-to-video generation model.

To address the generation quality issues caused by relying on a single coarse condition, we introduce multiple fine-grained conditions as intermediaries. Specifically, we utilize fine-grained poses and 3D hands. Fine-grained poses capture the signer's body posture and facial expressions, while 3D hands provide detailed and accurate descriptions of hand movements, even in the presence of occlusions. To avoid directly translating error-prone high-dimensional conditions, SignViP employs a discrete tokenization paradigm with effective integration and representation of these fine-grained conditions.

As shown in Figure 2, the spoken language text $\mathcal{T}$ is first translated into the discrete multi-condition tokens $\mathbf{d}_{0:F-1}$ by the **Multi-Condition Token Translator**. The multi-condition tokens $\mathbf{d}_{0:F-1}$ are then decoded by the **FSQ Autoencoder** to continuous multi-condition embeddings $\mathbf{e}_{0:F-1}$, which are equivalent to the embeddings obtained from a multi-condition encoder that encodes multiple fine-grained conditions (*i.e.*, fine-grained poses and 3D hands). Finally, $\mathbf{e}_{0:F-1}$ are injected into **Sign Video Diffusion Model** to guide the sign language video generation (*i.e.*, animating the signer in reference image $\mathbf{x}_\star$). The overall pipeline of SignViP can be formulated as

$$p_\theta(\mathbf{X}|\mathbf{x}_\star, \mathcal{T}) = p_{\theta_{\text{gen}}}(\mathbf{X}|\mathbf{x}_\star, \mathbf{e}_{0:F-1}) \cdot p_{\theta_{\text{AE}}}(\mathbf{e}_{0:F-1}|\mathbf{d}_{0:F-1}) \cdot p_{\theta_{\text{tran}}}(\mathbf{d}_{0:F-1}|\mathcal{T}) , \qquad (4)$$

where $\theta_{\text{gen}}$, $\theta_{\text{AE}}$, and $\theta_{\text{tran}}$ denote parameters of Sign Video Diffusion Model, FSQ Autoencoder, and Multi-Condition Token Translator, respectively.

### 3.3 Construction of Multi-Condition Token Space

SignViP is trained with three steps to construct the multi-condition token space for the discrete tokenization paradigm.

**Step I.** We train Sign Video Diffusion Model with a multi-condition encoder (*i.e.*, a multi-layer convolution network) using a denoising loss to establish a connection between the conditions and the sign language videos. Specifically, multiple fine-grained conditions (*e.g.*, fine-grained poses and 3D hands) are encoded by the multi-condition encoder into the continuous multi-condition embeddings $\mathbf{e}_{0:F-1}$. These embeddings, along with the reference image $\mathbf{x}_\star$, serve as the guidance signals for Sign Video Diffusion Model to perform diffusion process. More details can be found in Section 3.4.

**Step II.** We train FSQ Autoencoder using a reconstruction loss to learn the compression and quantization of multi-condition embeddings $\mathbf{e}_{0:F-1}$. The encoder $\mathcal{E}_{\text{FSQ}}$ of FSQ Autoencoder compresses and quantizes the embeddings $\mathbf{e}_{0:F-1}$ into discrete multi-condition tokens $\mathbf{d}_{0:F-1}$, while the its decoder $\mathcal{D}_{\text{FSQ}}$ reconstructs $\mathbf{e}_{0:F-1}$ from $\mathbf{d}_{0:F-1}$. More details can be found in Section 3.5.

**Step III.** We train the Multi-Condition Token Translator to autoregressively translate spoken language text $\mathcal{T}$ to the multi-condition tokens $\mathbf{d}_{0:F-1}$. More details can be found in Section 3.6.

### 3.4 Sign Video Diffusion Model

Sign Video Diffusion Model aims to generate sign language videos in a diffusion-based manner [70, 28] under the guidance of the reference image $\mathbf{x}_\star$ and the continuous multi-condition embeddings $\mathbf{e}_{0:F-1}$. Inspired by the previous works [29, 91], as shown in Figure 2(3), Sign Video Diffusion Model consists of three modules: Condition Guider, Denoising U-Net, and Reference Net. Condition Guider and Reference Net respectively encode the multi-condition embeddings $\mathbf{e}_{0:F-1}$ and the reference image $\mathbf{x}_\star$ to guide the Denoising U-Net.

**Denoising U-Net** is the backbone of the Sign Video Diffusion Model, which mirrors the architecture of Stable Diffusion (SD) v1.5 [55]. Each U-Net block includes a ResNet layer [22], a self-attention layer, and a temporal-attention layer [21]. The self-attention layer and the temporal-attention layer perform attention operation [76] along the spatial axes and the temporal axis, respectively.

**Condition Guider** is a lightweight guidance network that encodes $\mathbf{e}_{0:F-1}$, whose each block consists of convolution layers and a temporal attention layer. The output feature of each block is added to the corresponding block's feature in the downsampling part of the Denoising U-Net.

**Reference Net** [29] shares the same architecture of SDv1.5 and operates in parallel with the Denoising U-Net. The reference image $\mathbf{x}_\star$ is first encoded into the latent space by the VAE encoder $\mathcal{E}_{\text{VAE}}$,

$\mathbf{z}_\star = \mathcal{E}_{\mathrm{VAE}}(\mathbf{x}_\star)$. The encoded reference latent $\mathbf{z}_\star$ is then fed into the Reference Net. The output feature of the self-attention layer in each block of the Reference Net is spatially concatenated with the input feature of the self-attention layer in the corresponding block of the Denoising U-Net.

During training, the loss function is the extended denoising loss of Equation 1,

$$\mathcal{L}_{\mathrm{denoise}} = \mathbb{E}_{\mathbf{Z}_0, \boldsymbol{\epsilon} \sim \mathcal{N}(\mathbf{0}, \boldsymbol{I}), t} \| \boldsymbol{\epsilon}_\theta(\mathbf{Z}_t; \mathbf{r}_\star, \mathbf{E}, t) - \boldsymbol{\epsilon}_t \|^2 . \tag{5}$$

Among them, $\mathbf{Z}_0 = \mathcal{E}_{\mathrm{VAE}}(\mathbf{X}_0)$ is the target latent which is encoded from the target video $\mathbf{X}_0$. $\boldsymbol{\epsilon}_\theta$ is the Denoising U-Net. $\mathbf{r}_\star = \mathcal{R}(\mathbf{z}_\star)$ is the reference features, which are encoded by the Reference Net $\mathcal{R}$. $\mathbf{E} = \mathcal{C}(\mathbf{e}_{0:F-1})$ is the conditional features, which are encoded by the Condition Guider $\mathcal{C}$.

Considering that subtle pose variations in sign language videos carry important semantic meaning, the model needs to be robust to potential anomalies in the generated condition sequences. To address this, we propose *condition augmentation*. Specifically, each condition frame has a probability $p$ of being randomly replaced with frames from other videos, deliberately introducing controlled disruptions in the temporal continuity. By exposing the model to these artificial discontinuities during training, we effectively enhance its robustness to unexpected conditional transitions.

The inference starts from the sampled Gaussian noise. Then, the diffusion scheduler (*e.g.*, DDIM [71]) is applied to generate images with multiple denoising steps. For each inference step, the noise prediction relies on Classifier-Free Guidance (CFG) [27]. Finally, the generated video is achieved from the latent by a VAE decoder $\mathcal{D}_{\mathrm{VAE}}$.

## 3.5 FSQ Autoencoder

The FSQ Autoencoder is designed to establish a connection between the multi-condition embeddings $\mathbf{e}_{0:F-1}$ and the corresponding discrete tokens $\mathbf{d}_{0:F-1}$. The pre-trained multi-condition encoder first encodes multiple conditions to the continuous embeddings $\mathbf{e}_{0:F-1}$. These embeddings are subsequently compressed and quantized into $\mathbf{d}_{0:F-1}$ by the FSQ Autoencoder encoder $\mathcal{E}_{\mathrm{FSQ}}$, which provides a compact representation for the Multi-Condition Token Translator. Finally, the FSQ Autoencoder decoder $\mathcal{D}_{\mathrm{FSQ}}$ dequantizes $\mathbf{d}_{0:F-1}$ and reconstructs $\mathbf{e}_{0:F-1}$. The training objective of the FSQ Autoencoder is an L2 reconstruction loss.

$$\mathcal{L}_{\mathrm{FSQ\text{-}AE}} = \mathbb{E}_{\mathbf{e}_{0:F-1}} \| \mathcal{D}_{\mathrm{FSQ}}(\mathcal{E}_{\mathrm{FSQ}}(\mathbf{e}_{0:F-1})) - \mathbf{e}_{0:F-1} \|^2 . \tag{6}$$

The architecture of the FSQ Autoencoder follows that of the VAE. Instead of applying variational Bayesian inference in the latent space, it performs the FSQ operation, as illustrated in Equation 2.

## 3.6 Multi-Condition Token Translator

Multi-Condition Token Translator is designed to translate the spoken language text $\mathcal{T}$ into the discrete multi-condition tokens $\mathbf{d}_{0:F-1}$. Since sign language videos often exceed 100 frames, and each frame should maintain coherent temporal relationships without a strict internal order, we design a frame-level autoregressive model.

As shown in Figure 2(1), following previous works [86, 3], the spoken language text $\mathcal{T}$ is firstly encoded by CLIP text encoder [49] to obtain semantic embeddings, which serve as the initial input hidden states of the GPT-2 model [48]. Each output hidden state of the GPT model is decoded through multiple parallel prediction heads to generate all tokens of the corresponding frame simultaneously. On the input side, tokens belonging to the same frame are mixed to obtain unified input hidden states. Unlike methods that require dedicated modules for video length prediction [83] or rely on real video lengths [58], Multi-Condition Token Translator naturally determines the video generation endpoint by producing an "[EOS]" token.

During training, given the pre-trained multi-condition encoder and FSQ Autoencoder encoder, the produced tokens are considered the ground-truth. The cross-entropy loss is computed between the predicted tokens $\hat{\mathbf{b}}_{0:F}$ and the ground-truth tokens $\mathbf{b}_{0:F}$[3],

$$\mathcal{L}_{\mathrm{AR}} = \frac{1}{F+1} \sum_{i=0}^{F} \mathrm{CrossEntropy}(\hat{\mathbf{b}}_i; \mathbf{b}_i) . \tag{7}$$

---

[3]Note that we use $F$ instead of $F-1$ here due to the inclusion of the additional endpoint token "[EOS]".

To mitigate the exposure bias issue between training and inference, we employ a *scheduled sampling strategy*, wherein 40% of the input tokens are randomly replaced with arbitrary indices from the vocabulary during training. This approach improves the model's robustness and generalization performance during inference.

# 4 Experiments

## 4.1 Experimental Settings

**Datasets.** We employ two sign language datasets for experiments. (1) *RWTH-2014T* [5] is a German sign language dataset. It comprises 8,257 sign language videos. The dataset is divided into 7,096 training samples, 519 validation samples, and 642 test samples. To align with the $8\times$ downsampling rate of VAE, the frame size was resized from $260\times210$ to $272\times224$. (2) *How2Sign* [11] is an American sign language dataset. It includes 2,456 sign language videos. Using the provided timestamps, we segmented the videos to create a sentence-level dataset. This dataset consists of 31,128 training samples, 2,322 test samples, and 1,741 validation samples. The frame size is set to $512\times512$.

**Evaluation Metrics.** To evaluate semantic consistency, we utilize the back-translation metrics following ProTran [58]. Specifically, we train an SLT model [6] to translate sign language videos or poses back into texts. The back-translated text is then compared with the ground-truth text with metrics of *BLEU* [44] and *ROUGE-L* [34]. To provide a more comprehensive evaluation of back-translated texts, we further employ the *COMET* [52, 51] metric, which is specifically designed to predict human judgments of machine translation quality. COMET is widely used for machine translation tasks [19, 1, 24] and is considered more suitable than BLEU and ROUGE. To evaluate the video quality, we employ *FID* [26], *CLIP-FID*, *FVD* [74], and *Identity Similarity (IDS)*. Among them, CLIP-FID is a variation of FID that utilizes CLIP [49] embedding as the frame's embedding. IDS measures the identity consistency between generated and ground-truth videos. It calculates the cosine similarity of face embeddings extracted using YOLO5Face [46] and Arc2Face [43]. To further investigate the generative capability of video diffusion models in addition to *FVD* [74], we employ frame-level metrics including *PSNR* [13], *SSIM* [81], and *LPIPS* [89], leveraging their suitability in scenarios where ground-truth videos are temporally aligned with the generated videos. Additionally, we introduce *Hand SSIM*, which measures SSIM specifically in the hand region for a more precise evaluation of the hand quality.

**Implementation Details.** In *Multi-Condition Token Translator*, we utilize a multilingual version of the CLIP model[4] to enable handling of multiple spoken language texts effectively. In *FSQ Autoencoder*, the encoder and decoder follow the architecture of their counterparts in VAE. Specifically, FSQ Autoencoder applies 4 latent channels, with each channel having a quantization level of 5. Together, this results in a total vocabulary size of 625, computed as $5^4 = 625$ due to the combination of levels across all channels. In *Sign Video Diffusion Model*, both the Denoising U-Net and the Reference Net are initialized with Stable Diffusion v1.5[5]. The temporal-attention layers in the Denoising U-Net are initialized from AnimateDiff [21]. The condition augmentation rate is set to 0.001. During inference, Sign Video Diffusion Model utilizes a guidance scale of 3.5 for CFG. Additionally, the number of inference steps is configured to 50.

**Training Details.** The training of the three stages are conducted on 4 NVIDIA RTX A6000 GPUs using Adam optimizer [30], with each stage consisting of 50,000 training steps. The batch sizes of stage I, II, and III are 2, 16, and 16. Their learning rates are 1e-5, 5e-5, and 1e-6.

## 4.2 Comparison

**Back-Translation Comparison.** To quantitatively evaluate the semantic accuracy of the generated sign language videos, we perform two types of back-translation comparisons. Specifically, we respectively train a video-to-text translation model and a pose-to-text translation model to compare with SLVG methods and SLP methods [6]. For pose back-translation comparison, we extract pose sequences from the generated videos using OpenPose [7] and a 2D-to-3D mapping method [87]. As

---

[4] https://huggingface.co/sentence-transformers/clip-ViT-B-32-multilingual-v1
[5] https://huggingface.co/stable-diffusion-v1-5/stable-diffusion-v1-5

Table 1: Comparison of **video back-translation** performance.

| | RWTH-2014T | | | | | | How2Sign | | | | | |
|---|---|---|---|---|---|---|---|---|---|---|---|---|
| | BLEU-1 | BLEU-2 | BLEU-3 | BLEU-4 | ROUGE | COMET | BLEU-1 | BLEU-2 | BLEU-3 | BLEU-4 | ROUGE | COMET |
| Ground-Truth | 33.06 | 20.81 | 15.00 | 11.90 | 34.27 | 0.6157 | 20.37 | 13.11 | 9.78 | 7.53 | 21.43 | 0.5882 |
| MoMP [59] + ControlNet [88] | 19.12 | 8.95 | 5.33 | 3.61 | 21.54 | 0.5033 | 12.53 | 5.59 | 3.48 | 2.31 | 13.72 | 0.5122 |
| MoMP [59] + AnimateAnyone [29] | 20.05 | 8.79 | 5.24 | 3.72 | 21.68 | 0.5091 | 13.65 | 5.82 | 3.39 | 2.25 | 14.15 | 0.5208 |
| SignGAN [61] | 17.41 | 7.93 | 4.67 | 3.16 | 19.64 | 0.4977 | 10.66 | 4.62 | 2.92 | 1.97 | 11.76 | 0.5104 |
| *w/* AnimateAnyone [29] | 18.29 | 7.75 | 4.59 | 3.23 | 19.70 | 0.4928 | 11.82 | 4.85 | 2.83 | 1.92 | 12.12 | 0.5135 |
| SignGen [47] | 13.28 | 3.05 | 1.13 | 0.51 | 16.13 | 0.4086 | 8.21 | 1.91 | 0.64 | 0.41 | 9.54 | 0.4127 |
| **SignViP (Ours)** | **26.72** | **15.65** | **11.14** | **8.65** | **28.85** | **0.5608** | **16.21** | **9.36** | **6.28** | **5.04** | **16.99** | **0.5524** |

Table 2: Comparison of **pose back-translation** performance.

| | RWTH-2014T | | | | | | How2Sign | | | | | |
|---|---|---|---|---|---|---|---|---|---|---|---|---|
| | BLEU-1 | BLEU-2 | BLEU-3 | BLEU-4 | ROUGE | COMET | BLEU-1 | BLEU-2 | BLEU-3 | BLEU-4 | ROUGE | COMET |
| Ground-Truth | 30.99 | 18.36 | 12.83 | 9.87 | 31.02 | 0.5978 | 24.56 | 14.96 | 10.31 | 7.91 | 24.88 | 0.6250 |
| ProTran [58] | 17.96 | 8.99 | 5.64 | 4.07 | 20.97 | 0.5091 | 14.57 | 7.47 | 4.59 | 3.42 | 17.32 | 0.5549 |
| Adversarial [56] | 17.70 | 8.96 | 5.72 | 4.18 | 21.15 | 0.5127 | 14.76 | 7.15 | 4.66 | 3.48 | 17.84 | 0.5618 |
| MDN [60] | 18.06 | 9.30 | 6.06 | 4.52 | 21.44 | 0.5251 | 14.94 | 7.54 | 5.10 | 3.67 | 18.21 | 0.5685 |
| MoMP [59] | 20.55 | **10.98** | **7.02** | **5.14** | **23.75** | **0.5466** | 16.57 | **8.47** | 5.38 | 4.16 | **19.38** | **0.5802** |
| SignGAN [61] | 12.14 | 6.10 | 3.88 | 2.85 | 14.79 | 0.5123 | 10.86 | 5.27 | 3.30 | 2.62 | 13.19 | 0.5673 |
| *w/* AnimateAnyone [29] | 12.36 | 6.23 | 4.01 | 2.97 | 14.93 | 0.5231 | 10.97 | 5.36 | 3.39 | 2.67 | 13.36 | 0.5315 |
| SignGen [47] | 10.42 | 2.42 | 0.89 | 0.38 | 12.68 | 0.4324 | 8.67 | 1.79 | 2.41 | 1.36 | 8.69 | 0.4413 |
| **SignViP (Ours)** | **21.94** | 10.06 | 6.32 | 4.61 | 22.67 | 0.5347 | **17.35** | 8.28 | **5.41** | **4.42** | 18.23 | 0.5738 |

shown in Table 1, in *video back-translation comparison*, SignViP outperforms all competing methods, including SignGAN [61], its enhanced version using AnimateAnyone [29], and SignGen [47]. These results validate SignViP as a more reliable solution for SLVG by effectively preserving semantic consistency. As shown in Table 2, in *pose back-translation comparison*, SignViP consistently outperforms previous SLVG methods and SLP methods (*i.e.*, ProTran [58], Adversarial Training [56], and MDN [60]) across most evaluation metrics. Although MoMP [59] achieves slightly higher scores than our SignViP on certain metrics, our method remains highly competitive overall. It is worth noting that these SLP baselines translate text directly into pose sequences, which aligns with our pose back-translation evaluation pipeline. In contrast, our SLVG method requires detecting poses from the generated videos, potentially introducing additional errors that could impact evaluation results. To enable a more fair comparison under the SLVG setting, we further combine the state-of-the-art SLP method, MoMP, with a pose-to-video generation approach (*i.e.*, ControlNet [88] or AnimateAnyone [29]). As shown in the first two rows of Table 1, the video back-translation results demonstrate that our method is better suited for the SLVG task compared to MoMP-based methods. These results further underscore SignViP's effectiveness in preserving semantic accuracy.

**Video Quality Comparison.** Table 3 summarizes the video quality comparison of the generated sign language videos. Our proposed SignViP method significantly outperforms prior SLVG approaches across all evaluated metrics. Specifically, the lowest FID, CLIP-FID, and FVD achieved by our model demonstrate its superior ability to generate sign language videos that are not only visually realistic but also exhibit high temporal coherence and natural motion consistency. Furthermore, the highest IDS scores achieved by our method highlight its effectiveness in accurately preserving the identity of the signer. These results collectively validate the efficacy of SignViP in producing high-fidelity, visually coherent, and perceptually realistic sign language videos.

**Generative Capability Comparison for Video Diffusion Models.** To compare the generative capabilities of different video diffusion models, we evaluate three methods, which are ControlNet [88], AnimateAnyone [29], and our Sign Video Diffusion Model. As detailed in Table 4, our Sign Video Diffusion Model consistently outperforms other methods across all metrics. Specifically, our Hand SSIM outperforms others, highlighting our model's ability to preserve hand details. The results clearly highlight the superiority of our method.

**Qualitative Comparison.** We present the qualitative results in Figure 3(a) of the previous SLVG methods and our SignViP. Compared to the previous methods, SignViP generates higher-quality sign language videos while maintaining greater semantic accuracy with the spoken language text.

Table 3: Comparison of video quality.

| | RWTH-2014T | | | | How2Sign | | | |
|---|---|---|---|---|---|---|---|---|
| | FID ↓ | CLIP-FID ↓ | FVD ↓ | IDS ↑ | FID ↓ | CLIP-FID ↓ | FVD ↓ | IDS ↑ |
| SignGAN [61] | 547.90 | 167.70 | 1431.38 | 0.463 | 667.44 | 210.11 | 2766.97 | 0.538 |
| *w/* AnimateAnyone [29] | 595.99 | 161.97 | 1330.54 | 0.462 | 679.41 | 215.05 | 2484.39 | 0.533 |
| SignGen [47] | 644.06 | 184.66 | 1715.32 | 0.515 | 815.69 | 186.32 | 3538.49 | 0.539 |
| **SignViP (Ours)** | **508.91** | **154.10** | **1025.45** | **0.571** | **575.67** | **109.61** | **2207.67** | **0.624** |

Table 4: Generative capability comparison of video diffusion models.

| | RWTH-2014T | | | | | How2Sign | | | | |
|---|---|---|---|---|---|---|---|---|---|---|
| | FVD ↓ | SSIM ↑ | PSNR ↑ | LPIPS ↓ | Hand SSIM ↑ | FVD ↓ | SSIM ↑ | PSNR ↑ | LPIPS ↓ | Hand SSIM ↑ |
| ControlNet [88] | 556.63 | 0.784 | 19.50 | 0.137 | 0.483 | 427.22 | 0.826 | 21.32 | 0.116 | 0.657 |
| AnimateAnyone [29] | 365.42 | 0.794 | 20.06 | 0.121 | 0.505 | 293.18 | 0.821 | 21.54 | 0.103 | 0.663 |
| **Sign Video Diffusion Model (Ours)** | **275.22** | **0.829** | **22.91** | **0.089** | **0.614** | **210.63** | **0.855** | **23.11** | **0.074** | **0.752** |

## 4.3 Model Study

**Identity Generalization.** Figure 3(b) showcases how our SignViP generalizes signer identities by adapting appearance guidance from distinct reference images. This demonstrates the robustness of our method in preserving signer-specific appearances while ensuring accurate sign language translation.

**Effect of Multiple Conditions.** To evaluate whether incorporating multiple fine-grained conditions improves video quality, we ablate the fine-grained poses and 3D hands from our pipeline, respectively. As shown in Table 5, removing one of the conditions leads to a substantial performance degradation across

Table 5: Effect of multiple conditions.

| | FVD | Hand SSIM | BLEU-4 | ROUGE-L |
|---|---|---|---|---|
| *w/o* 3D Hands | 382.64 | 0.477 | 5.39 | 23.98 |
| *w/o* Fine-Grained Poses | 461.98 | 0.488 | 3.13 | 19.41 |
| **Multiple Conditions** | **275.22** | **0.614** | **8.65** | **28.85** |

all metrics. These results demonstrate that incorporating multiple fine-grained conditions is essential for enhancing both the semantic accuracy and visual quality of the generated videos.

**Effect of Compression.** To investigate the necessity of compression for SignViP, we conduct experiments to assess the impact of the compression/downsampling rate in the FSQ Autoencoder. As illustrated in Figure 4(a), the performance of back-translation improves notably as the compression rate increases. Notably, when the compression rate is set to 1 (*i.e.*, no compression is applied), the model demonstrates significantly poor performance. These results underscore the critical role of compression in enhancing the effectiveness of SignViP.

**Effect of Quantization.** To investigate the necessity of quantization for SignViP, we conduct an experiment where FSQ is not performed during FSQ Autoencoder, while Multi-Condition Token Translator is trained with continuous embedding prediction. As illustrated in Figure 3(c), we observed that continuous embedding prediction poses significant challenges for the translator, resulting in weak semantic alignment and low video quality. When incorporating FSQ, we achieve substantial improved performance. These findings highlight the importance of quantization for our SignViP.

**Effect of Condition Augmentation.** To evaluate the impact of condition augmentation (Section 3.4) on generation quality, we conducted experiments by varying the augmentation probability $p$. Figure 4(b) presents the results of condition augmentation with varying values of $p$. Specifically, introducing a small probability of augmentation (*i.e.*, $p = 10^{-3}$) slightly improves FVD and ROUGE-L scores, suggesting enhanced video quality and linguistic consistency. However, as $p$ increases further, the effectiveness of condition augmentation diminishes. These results indicate that excessive augmentation introduces too much randomness, impacting both video quality and textual coherence.

**Effect of Scheduled Sampling Strategy.** To evaluate the effect of varying the sampling ratio $r$ of the scheduled sampling strategy (Section 3.6) on generation quality, we conducted experiments by varying $r$. The experimental results, as shown in Figure 4(c), reveal that the scheduled sampling strategy significantly impacts the quality of generated outputs. When $r = 1$, meaning all input tokens are replaced with random indices, the results indicate that excessive randomness severely hurts the model's consistency and coherence. As $r$ decreases, generation quality improves steadily. The best performance is observed at $r = 0.4$. However, as $r$ is further reduced to 0.2, performance begins to

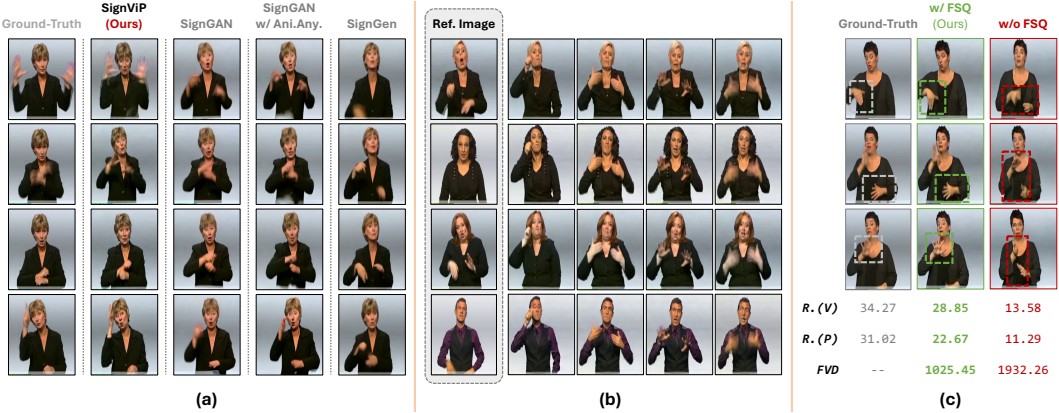

Figure 3: (a) Qualitative comparison on RWTH-2014T dataset. (b) Visual examples illustrating our SignViP's capability for identity generalization. (c) Effect of quantization. "R.(V)" and "R.(P)" mean ROUGE metrics of video and pose back-translation, as shown in Table 1 and Table 2, respectively. "FVD" evaluates the protocol described in Table 3.

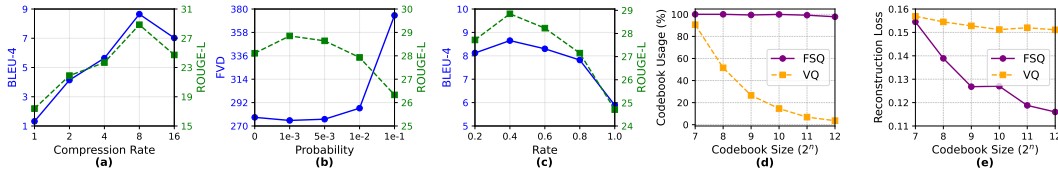

Figure 4: (a) Effect of compression rate. (b) Effect of condition augmentation probability. Note that "FVD" evaluates the protocol described in Table 4. (c) Effect of sampling rate for the scheduled sampling strategy. (d) Codebook usage comparison between FSQ and VQ. (e) Multi-conditional reconstruction loss comparison between FSQ and VQ.

decline slightly. This suggests that a very low replacement ratio is insufficient to simulate the diverse distributions encountered during inference, leading to suboptimal performance.

**FSQ vs. VQ.** To compare the performance of FSQ with traditional Vector Quantization (VQ), we evaluate both methods in terms of codebook usage efficiency and multi-conditional reconstruction loss. *(1) Codebook Usage.* To assess the efficiency of codebook utilization, we conducted experiments with varying codebook sizes ranging from $2^7$ to $2^{12}$, and the results are summarized in Figure 4(d). The results demonstrate that FSQ consistently achieves high codebook usage rates, remaining above 97% even with larger codebook. In contrast, VQ experiences a sharp decrease in usage as the codebook size increases. These findings highlight the stability and scalability of FSQ. *(2) Reconstruction Loss.* To evaluate the ability of conditional preservation, we measured the reconstruction loss of both methods under different codebook sizes, as detailed in Figure 4(e). The results show that FSQ achieves consistently lower reconstruction loss compared to VQ, demonstrating its superior capability in preserving original conditional structure.

## 5 Conclusion

In this work, we propose SignViP, a novel Sign Language Video Generation (SLVG) framework that incorporates multiple fine-grained conditions to enhance generation fidelity by adopting a discrete tokenization paradigm. SignViP consists of three components: (1) Sign Video Diffusion Model, which learns continuous embeddings encapsulating fine-grained motion and appearance details, (2) FSQ Autoencoder, which compresses and quantizes these embeddings into discrete tokens for compact representation, and (3) Multi-Condition Token Translator, which translates spoken language text to discrete multi-condition tokens. Experimental results demonstrate that SignViP achieves state-of-the-art performance in video quality, temporal coherence, and semantic fidelity.

## Acknowledgments

We would like to thank the anonymous reviewers for their insightful comments. This work is supported by the JiangSu Natural Science Foundation under Grant No. BK20251989; the National Natural Science Foundation of China under Grants Nos. 62172208, 62441225, 61972192; the Fundamental Research Funds for the Central Universities under Grant No. 14380001. This work is partially supported by Collaborative Innovation Center of Novel Software Technology and Industrialization.

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

## A  Limitations

In this paper, we propose a novel SLVG framework, SignViP, which demonstrates significant improvements over previous methods. Nevertheless, our framework still has two notable limitations that warrant further investigation.

The first limitation is its inability to support multiple sign languages simultaneously. As different countries have their own unique sign language systems, adapting SignViP for a specific sign language currently requires training a separate model for each. This dependency on independent model training significantly restricts its practicality in real-world applications. Consequently, the development of a unified SLVG system capable of supporting multiple sign languages will be critical for enhancing its versatility and applicability. Addressing this challenge will serve as a key direction in our future research endeavors.

The second limitation pertains to the computational inefficiency inherent in the iterative denoising process of diffusion models. Existing methods, such as the consistency model [72] and efficient ODE solvers [36, 37], offer promising approaches to accelerate the sampling process of diffusion-based models. Incorporating these techniques into SignViP represents another avenue for improving its efficiency and scalability, which will also be prioritized in our future research.

## B  Architecture Details

**Multi-Condition Encoder.** The multi-condition token space in our framework is constructed using a multi-condition encoder. This encoder adopts a simple convolutional architecture with 8 convolutional layers with SiLU activation [12, 50], each achieving a $4\times$ spatial downsampling (*i.e.*, $2\times$ along both the height and width axes).

**Mixer in Multi-Condition Token Translator.** In Multi-Condition Token Translator, we use a mixer to aggregate tokens from the same frame into unified input hidden states. The mixer consists of a linear layer, followed by LayerNorm and GELU activation [25]. Tokens are first projected into embeddings, which are then flattened and passed through the mixer to produce a single hidden state for each frame.

**Decoder in Multi-Condition Token Translator.** In the Multi-Condition Token Translator, we utilize a decoder to decode all tokens of the same frame from the output hidden states of the GPT-2 model. The decoder employs multiple parallel heads to decode each token within the frame. Specifically, each head is implemented as a lightweight Transformer layer [76, 8, 15, 16].

## C  Training Details of Back-Translation Models

To evaluate the semantic consistency of the generated sign language videos, we follow ProTran [58] to train two SLP models [6] to translate sign language videos (*i.e.*, the video back-translation model) and poses (*i.e.*, the pose back-translation model) into texts, respectively.

**Video Back-Translation Model.** The video back-translation model is trained using a single NVIDIA RTX A6000 GPU. An Adam optimizer [30] is utilized with a learning rate of 1e-3, and the batch size is set to 32. Validation performance is logged every 5 steps, providing checkpoints throughout the training process. The best-performing model checkpoint is observed at step 10,500.

**Pose Back-Translation Model.** The pose back-translation model is also trained on a single NVIDIA RTX A6000 GPU. The training configuration mirrors that of the video back-translation model, using an Adam optimizer [30] with a learning rate of 1e-3, a batch size of 32, and validation logged every 5 steps. The optimal model checkpoint is reached at step 5,400.

## D  Human Evaluation

The evaluation protocol based on back-translation models is highly dependent on the quality of the back-translation system, which may introduce certain biases. Although human evaluation could potentially offer a more reliable solution, recruiting qualified sign language experts presents significant challenges.

To address this limitation, we employ a compromise human evaluation strategy that does not require expert knowledge of sign language. Specifically, we present human evaluators with the ground-truth sign language video, along with several anonymized candidate videos generated by different SLVG methods. Evaluators are instructed as follows: "*Please choose the video where the signer's actions appear most similar to those in the ground-truth video.*"

For this evaluation, we recruited 10 non-expert participants. Each participant evaluated 25 groups of samples for the RWTH-2014T dataset and 50 groups for the How2Sign dataset, resulting in a total of 250 and 500 votes, respectively. We report the proportion of votes received by each SLVG method in Table 6. Notably, our method, SignViP, achieves the highest vote proportion across both datasets, indicating that SignViP generates sign language videos that are more consistent with the ground-truth references as perceived by human evaluators.

Table 6: Human Evaluation Results on RWTH-2014T and How2Sign datasets.

|  | RWTH-2014T | | How2Sign | |
|---|---|---|---|---|
|  | #Votes | Vote% | #Votes | Vote% |
| *TOTAL* | *250* | *100.0%* | *500* | *100%* |
| SignGAN | 25 | 10.0% | 31 | 6.2% |
| *w/* AnimateAnyone | 23 | 9.2% | 24 | 4.8% |
| SignGen | 27 | 10.8% | 36 | 7.2% |
| SignViP (Ours) | **175** | **70.0%** | **409** | **81.8%** |

# E    More Experiments

## E.1    Comparison with Direct Condition Prediction

As stated in Section 1, one of the motivations for introducing the multi-condition token space is the inherent difficulty of directly translating fine-grained attributes. In Figure 1(4), we illustrate examples of direct multi-condition predictions, demonstrating significant discrepancies between ground-truth and predicted results. To quantitatively compare the direct condition prediction approach with our proposed method under the pose back-translation paradigm, we present experimental results in Table 7. The results demonstrate that direct condition prediction performs poorly across all metrics. These results clearly demonstrate that direct multi-condition prediction struggles to effectively model the fine-grained attributes and fails to maintain semantic consistency during the translation process.

Table 7: Comparison of pose back-translation performance between direct condition prediction and our proposed SignViP.

| Methods | RWTH-2014T | | | | | How2Sign | | | | |
|---|---|---|---|---|---|---|---|---|---|---|
|  | BLEU-1 | BLEU-2 | BLEU-3 | BLEU-4 | ROUGE | BLEU-1 | BLEU-2 | BLEU-3 | BLEU-4 | ROUGE |
| Ground-Truth | 30.99 | 18.36 | 12.83 | 9.87 | 31.02 | 24.56 | 14.96 | 10.31 | 7.91 | 24.88 |
| Condition Prediction | 9.27 | 2.23 | 0.80 | 0.32 | 11.64 | 5.39 | 1.63 | 2.26 | 1.13 | 8.12 |
| SignViP (Ours) | 21.94 | 10.06 | 6.32 | 4.61 | 22.67 | 17.35 | 8.28 | 5.41 | 4.42 | 18.23 |

## E.2    Efficiency Comparison

To compare the efficiency of our method with other approaches, we conducted experiments measuring the number of model parameters and inference time per frame. As shown in Table 8, the results demonstrate that our SignViP achieves comparable model size and inference time to the diffusion-based baselines. This indicates that our method delivers high-quality video generation without incurring significant additional computational cost, making it an efficient and practical solution for sign language video generation.

## E.3    Effectiveness of Pretrained Parameter Initialization

The comparison among methods is fair, as all the diffusion-based SLVG approaches (*i.e.*, Sign-GAN+AnimateAnyone, SignGen, and our SignViP) are initialized with parameters from existing pretrained diffusion models.

Table 8: Efficiency comparison between diffusion-based baselines and our SignViP.

|  | # Parameters | Inference Time (s/frame) |
|---|---|---|
| SignGAN *w/* AnimateAnyone [29] | 2575.85M | 1.3404 |
| SignGen | 2217.38M | 1.2225 |
| SignViP (Ours) | 2777.92M | 1.2398 |

Training a video generation diffusion model from scratch is extremely challenging due to the high computational costs and slow convergence. Therefore, leveraging pretrained diffusion parameters to initialize customized diffusion models has become a fundamental strategy to significantly accelerate training [23, 4, 53, 21, 39].

To further validate the effectiveness of pretrained parameter initialization, we conducted an ablation study, evaluating the quality of generated videos both with and without pretrained initialization under the same number of training steps. The results, presented in Table 9, clearly demonstrate the substantial advantage of initializing with pretrained parameters.

Table 9: Ablation study on the effect of pretrained initialization.

|  | FID ↓ | FVD ↓ |
|---|---|---|
| w/o Pretrained Initialization | 2278.23 | 3277.20 |
| **w/ Pretrained Initialization (Ours)** | **508.91** | **1025.45** |

## E.4 Order-Preserving Evaluation of Back-Translation Models

To further validate the reliability and comparability of our back-translation models, we conduct **order-preserving experiments**. Specifically, we introduce pose sequences with varying levels of errors and evaluate whether the corresponding output metrics display a consistent ranking that reflects the severity of the errors. In other words, a comparable back-translation model should exhibit a steady degradation in output quality as the degree of input error increases.

To simulate realistic pose errors, we independently apply spatial and temporal perturbations as follows: (1) **Spatial Perturbation**: Additive bias is applied to pose keypoints. To mimic real-world errors while avoiding excessive deformation, we first compute the variance of each keypoint's coordinates from the dataset. The bias added to each keypoint is sampled from a normal distribution $\mathcal{N}(0, \sigma^2)$, where $\sigma$ reflects the perturbation intensity. (2) **Temporal Perturbation**: We randomly delete, repeat, or duplicate pose frames at a ratio of $p$, where $p$ controls the perturbation intensity.

The results of the **pose back-translation model** under both spatial and temporal perturbations are summarized in Figure 5(a) and (b). As shown in the figures, all metrics decrease monotonically as the perturbation intensity increases. This demonstrates that our pose back-translation model is sensitive to varying levels of pose errors and can provide reliable evaluation metrics for comparison.

For the **video back-translation model**, we first synthesize videos from the perturbed poses using AnimateAnyone [29], and then evaluate the corresponding metrics. The results of the video back-translation model under both spatial and temporal perturbations are summarized in Figure 5(c) and (d). Consistent with our findings for the pose model, our video back-translation model also provides reliable evaluation metrics for comparison. Note that since we use AnimateAnyone to synthesize new videos, the metrics without perturbation differ from the ground-truth metrics reported in Table 1.

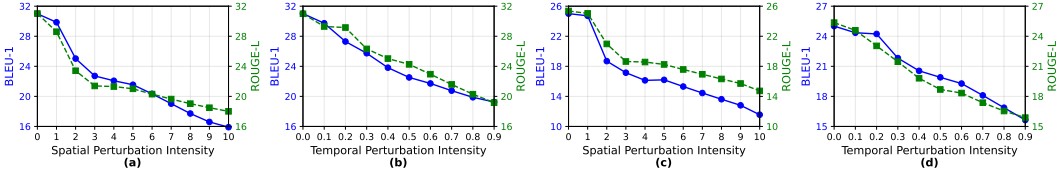

Figure 5: (a) Effect of spatial perturbation for the pose back-translation model. (b) Effect of temporal perturbation for the pose back-translation model. (c) Effect of spatial perturbation for the video back-translation model. (d) Effect of temporal perturbation for the video back-translation model.

# F More Cases

We demonstrate more video cases in Figure 6 and Figure 7.

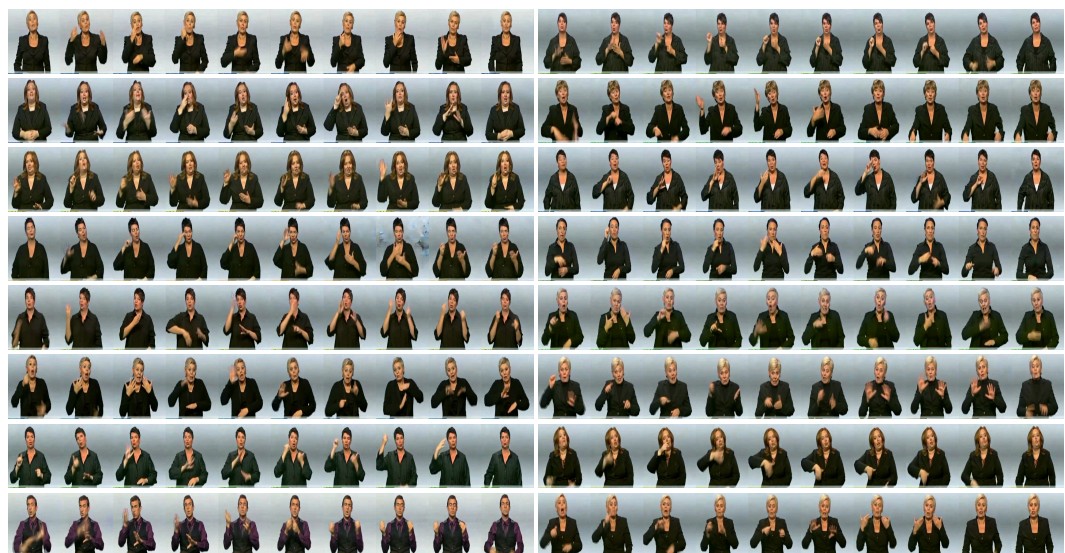

Figure 6: More generated cases of RWTH-2014T dataset.

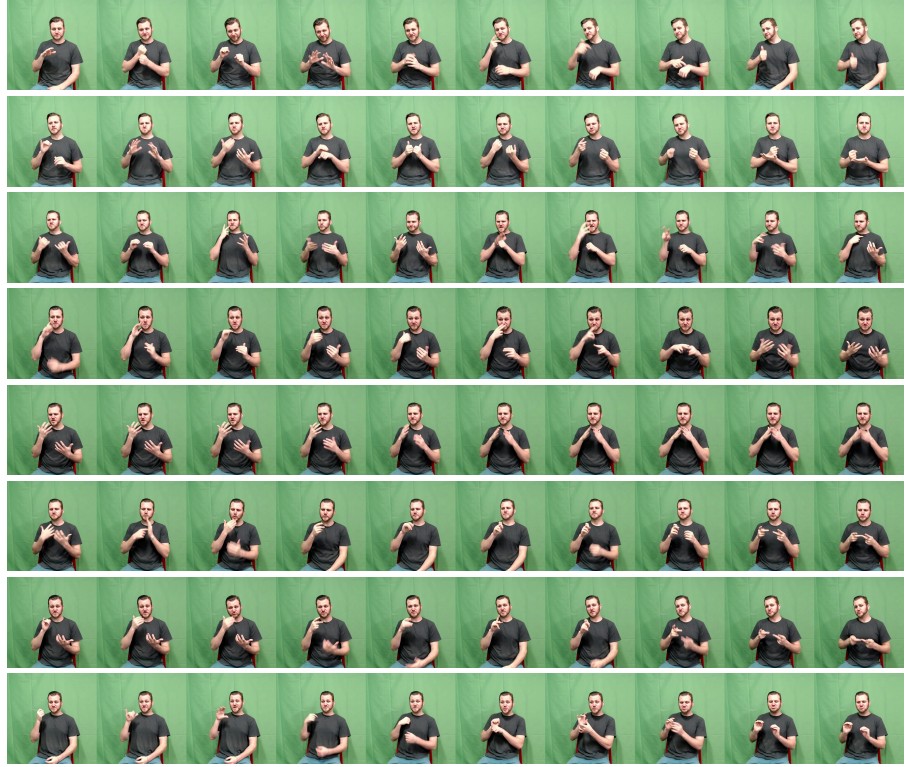

Figure 7: More generated cases of How2Sign dataset.

# G   Token Translation Accuracy

Directly computing the accuracy of token translation in our framework presents significant challenges. Due to temporal shifts and uneven scaling, translated tokens during inference may not be perfectly aligned with ground-truth tokens, making traditional accuracy metrics less reliable.

To address this, we employ **normalized Dynamic Time Warping (DTW) distance** as an alternative evaluation metric. DTW is a similarity measure that computes the optimal alignment between two sequences, even if they differ in length or are not aligned one-to-one. By normalizing the DTW distance, we accommodate variations in sequence length, enabling fair comparison across different settings. In our evaluation pipeline, we decode the translated tokens into condition embeddings using the decoder of the FSQ Autoencoder, and then compare these embeddings to ground-truth condition embeddings via normalized DTW distance.

To validate the effectiveness of normalized DTW distance as a metric, we conduct experiments following the settings described in "Effect of Compression" and "Effect of Scheduled Sampling Strategy" of Section 4.3. As shown in Tables 10 and 11, the trends of normalized DTW distance closely match those of previously reported metrics (BLEU-4 and ROUGE), confirming its reliability for evaluating token translation accuracy.

Table 10: Effect of Compression Rate on Token Translation Metrics

| Compression Rate | Norm. DTW Distance ($\downarrow$) | BLEU-4 | ROUGE |
| --- | --- | --- | --- |
| 1 | 1.9020 | 1.32 | 17.38 |
| 2 | 1.7937 | 4.14 | 21.89 |
| 4 | 1.7438 | 5.64 | 23.68 |
| **8 (Ours)** | **1.5968** | **8.65** | **28.85** |
| 16 | 1.7291 | 7.02 | 24.72 |

Table 11: Effect of Scheduled Sampling Rate on Token Translation Metrics

| Sampling Rate | Norm. DTW Distance ($\downarrow$) | BLEU-4 | ROUGE |
| --- | --- | --- | --- |
| 0.2 | 1.6123 | 8.12 | 27.72 |
| **0.4 (Ours)** | **1.5968** | **8.65** | **28.85** |
| 0.6 | 1.5978 | 8.30 | 28.23 |
| 0.8 | 1.6191 | 7.82 | 27.15 |
| 1.0 | 1.7354 | 5.89 | 24.71 |

