# OpenReview forum: "Advanced Sign Language Video Generation with Compressed and Quantized Multi-Condition Tokenization"
_NeurIPS.cc/2025/Conference — NeurIPS 2025 spotlight_

### Official Review · Reviewer_uzjH · 2025-06-19

**Clarity:** 3
**Significance:** 3
**Originality:** 3
**Rating:** 5
**Confidence:** 4

**Summary:**

The paper proposes SignViP, a novel Sign Language Video Generation method that incorporates Multiple Fine-Grained Conditions (poses and 3D hands) to enhance the quality of generated sign language videos. Their method includes three main components: 1) a sign video diffusion model, 2) a Finite Scalar Quantization autoencoder, and 3) a Multi-Condition Token Translator.

**Questions:**

**References**

Could use more references for SLVG? e.g., Pose-Guided Fine-Grained Sign Language Video Generation, ECCV 2024

**Questions**
- For video back-translation, you used sign language transformers [1] and [2] for pose back-translation, is that right?
- What is the resolution of the output video? Is it the same as input?
- In Figure 2, what is a Mixer? Please provide a short explanation for this figure, too. It is difficult to understand at first glance.
- In Section 3.6, the term mixed is vague. The authors didn't introduce what the mixer is, and it is found in the appendix. It is important to explain it in the paper and not the appendix.

[1] Necati Cihan Camgoz, Oscar Koller, Simon Hadfield, and Richard Bowden. Sign language transformers: Joint end-to-end sign language recognition and translation. In CVPR, 2020.

[2] Jan Zelinka and Jakub Kanis. Neural sign language synthesis: Words are our glosses. In WACV, 2020.

**Ethical Concerns:**

["NO or VERY MINOR ethics concerns only"]

**Final Justification:**

**final score**: The paper presents a clear engineering and robust method for sign language video generation, an area that needs more attention in the sign language domain. The method is presented clearly and is supported by thorough experiments. There were some minor issues from my side, and they have been addressed. Based on the rebuttal, I keep my score as **5: Accept**.

**Limitations:**

yes

**Paper Formatting Concerns:**

- In line 257, SPL is not defined before.

**Quality:**

3

**Strengths And Weaknesses:**

**Strengths**
- The paper presents strong comparative evaluations and ablation studies.
- Integrating the three main components demonstrates thoughtful engineering.
- The authors plan to release the code.

**Weaknesses**
- Since the training is not end-to-end and involves multiple separately trained stages, extending the pipeline to more datasets—especially in a multilingual setting—can significantly complicate the task.

---

> ### Author Rebuttal · Authors · 2025-07-31
>
> Thank you for recognizing the evaluations, engineering, and openness of our work. We will address your concerns below.
>
> # Weaknesses
>
> **`W1:`** Since the training involves multiple separately stages, extending the pipeline to more datasets—especially in a multilingual setting—can significantly complicate the task.
>
> **`R1:`** Although our SignViP framework consists of three training stages, only the final stage (Step III: Multi-Condition Token Translator) is directly language-dependent. The first and second stages are entirely language-agnostic, as they operate solely on videos and associated conditions without considering language categories.
>
> Consequently, when extending to multilingual datasets, the training of these two stages can be shared across languages. Specifically, the Sign Video Diffusion Model and FSQ Autoencoder can be pre-trained on large-scale human video datasets, even if these videos are not for sign language. Subsequently, for each target language, only the Multi-Condition Token Translator needs to be trained separately.
>
> Thus, our pipeline remains efficient and does not introduce significant complexity when scaling to multiple languages.
>
> # Questions
>
> **`Q1:`** Could you use more references for SLVG (e.g., PGMM [1])?
>
> **`A1:`** Thank you for pointing out the omission of one relevant work on SLVG. As this work has not released its code and pre-trained weights, we are unable to reproduce it from scratch and include it as a baseline in our experiments within the limited timeframe. However, we will reference and introduce this work in Section 2 (Related Works).
>
> ---
>
> **`Q2:`** Do you use Sign Language Transformers (SLT) [2] for video back-translation and the method from [3] for pose back-translation?
>
> **`A2:`** No. For both pose and video back-translation evaluations, we use SLT as the back-translation model. Specifically, for video back-translation, video frames are input into the SLT model. For pose back-translation, we follow the evaluation protocol of ProTran [4] and modify the original SLT architecture to accept pose points as input.
>
> ---
>
> **`Q3:`** What is the resolution of the output video? Is it the same as input?
>
> **`A3:`** For RWTH-2014T and How2Sign, the output resolutions are 272×224 and 512×512, respectively. The resolution of output video is the same as that of input. Although we have noted the resolution in Appendix B.1, we will include them in Section 4.1 to ensure clarity.
>
> ---
>
> **`Q4:`** In Figure 2, what is a Mixer?
>
> **`A4:`** The Mixer module is a lightweight MLP that aggregates tokens from the same frame into unified hidden states, which is described in Appendix C.
>
> ---
>
> **`Q5:`** Please provide a short explanation for Figure 2. It is difficult to understand at first glance.
>
> **`A5:`** Thanks for your suggestion. We will provide a detailed caption for Figure 2, as follows:
>
> "*Figure 2. Framework of our SignViP for sign language video generation (SLVG). (1) The spoken language text is translated into the multi-condition tokens by Multi-Condition Token Translator. (2) These tokens are decoded by FSQ Autoencoder into multi-condition embeddings, which are equivalent to the embeddings of multiple fine-grained conditions (i.e., fine-grained poses and 3D hands) generated by a multi-condition encoder. (3) The embeddings are injected into Sign Video Diffusion Model to guide the generation of sign language videos.*"
>
> ---
>
> **`Q6:`** In Section 3.6, the term "mixed" is vague. It is important to explain what the mixer is in the paper and not the appendix.
>
> **`A6:`** Thanks for your suggestion. We realize that not describing the structure and function of Mixer in the main text may cause confusion for readers, and we will include a detailed description of Mixer in the main part of the paper.
>
> # Paper Formatting Concerns
>
> **`C1:`** In line 257, "SPL" is not defined before.
>
> **`R1:`** Thank you for your reminder. This is a typographical error. It should be "SLP (Sign Language Production)" rather than "SPL".  We will correct this mistake.
>
> # References
>
> `[1]` Pose-Guided Fine-Grained Sign Language Video Generation, ECCV 2024.
> `[2]` Sign Language Transformers: Joint End-to-End Sign Language Recognition and Translation, CVPR 2020.
> `[3]` Neural Sign Language Synthesis: Words are Our Glosses. WACV 2020.
> `[4]` Progressive Transformers for End-to-End Sign Language Production, ECCV 2020.

---

> > ### Comment · Reviewer_uzjH · 2025-08-01
> >
> > I thank the authors for the clarification. My rating remains.

---

> > > ### Author Response · Authors · 2025-08-01
> > >
> > > We sincerely appreciate your recognition and are especially grateful for the positive rating.
> > >
> > > Thank you for taking the time to review our manuscript and responses, as well as for your insightful comments that will greatly help us improve our work.

---

### Official Review · Reviewer_8mrn · 2025-06-25

**Clarity:** 3
**Significance:** 3
**Originality:** 3
**Rating:** 5
**Confidence:** 3

**Summary:**

This paper introduces a new approach to generating sign language videos given a spoken language text, more specifically for American & German Sign Language. This is done through a multi-step process, leveraging multiple conditions to help improve the generation. Results showed strong performance in comparison to other recent approaches in this domain.

**Questions:**

You introduce Hand SSIM -- are you planning on releasing the details for this so others can leverage this?

**Ethical Concerns:**

["NO or VERY MINOR ethics concerns only"]

**Limitations:**

yes

**Paper Formatting Concerns:**

None.

**Quality:**

3

**Strengths And Weaknesses:**

Quality - the paper is technically sound for the most part. My biggest concern is the experiments, specifically the back translation. Sign language translation models still perform not too well in this domain, thus it makes it difficult to judge if the videos are generating correct sign language. Human evals would have strengthened the paper, though understandably it is difficult to find German/American Sign Language experts that can evaluate the results.

Clarity - the paper is well written and can be understood fairly well. Some areas of improvement though could be in the figures. The first one is a bit difficult to follow while reading the text, since you almost have 4 different figures lumped into one. And some of the figures can be very difficult to see (the ones with example video frames of signers) as they have been made so small.

Significance - These results are impactful, even with the concerns mentioned about about back translation in this domain. Sign language generation is a very difficult task that has seen more interest lately, and this paper does a good job at pushing the domain forward.

Originality -  the approach seems rather novel, incorporating new ideas to help it achieve better results than past approaches.

---

> ### Author Rebuttal · Authors · 2025-07-31
>
> We sincerely appreciate the recognition of the quality, clarity, significance, and originality of our work. We will address your concerns below.
>
> # Weaknesses
>
> **`W1:`** Sign language translation models still do not perform too well in this domain, thus it makes it difficult to judge if the videos are generating correct sign language. Human evaluations would have strengthened the paper, though understandably it is difficult to find sign language experts that can evaluate the results.
>
> **`R1:`** Thank you for your suggestion. To facilitate human evaluation for the SLVG task, we further adopt a compromise human evaluation strategy that does not require the sign language experts. Specifically, we present human evaluators with the ground-truth video alongside several corresponding anonymized videos generated by different methods. The evaluators are given the following instruction:
>
> "*Please choose the video where the signer's actions appear most similar to those in the ground-truth video.*"
>
> We invited 10 evaluators, each of whom received 25 group samples for the RWTH-2014T dataset and 50 for the How2Sign dataset, resulting in totals of 250 and 500 votes, respectively. We calculate the proportion of votes for each method, with the results shown below. Notably, our SignViP receives the highest proportion of votes, indicating that SignViP generates more accurate sign language videos.
>
> | | #Votes (RWTH) | Vote% (RWTH) | #Votes (H2S) | Vote% (H2S) |
> | :- | :-: | :-: | :-: | :-: |
> | SignGAN | 25 | 10.0% | 31 | 6.2% |
> | SignGAN w/ AnimateAnyone | 23 | 9.2% | 24 | 4.8% |
> | SignGen | 27 | 10.8% | 36 | 7.2% |
> | **SignViP (Ours)** | **175** | **70.0%** | **409** | **81.8%** |
> | *TOTAL* | *250* | *100.0%* | *500* | *100%* |
>
> ---
>
> **`W2:`** Some figures could be improved. (1) Figure 1 is a bit difficult to follow while reading the text, since you almost have 4 different figures lumped into one. (2) Some figures are very difficult to see (the ones with example video frames of signers) as they have been made so small.
>
> **`R2:`** Thank you for your suggestions. We will enhance the presentation of the figures. Specifically, Figure 1 will be reorganized to improve readability. In addition, all other figures containing video frames will be enlarged to make them easier for readers to view.
>
> # Questions
>
> **`Q1:`** Will you share the details of Hand SSIM to enable others to utilize it?
>
> **`A1:`** Yes, we are committed to sharing the implementation details of Hand SSIM by publicly releasing our code repository.

---

> > ### Comment · Reviewer_8mrn · 2025-08-04
> >
> > Thank you for addressing the concerns. My scores will remain as-is.

---

> > > ### Author Response · Authors · 2025-08-04
> > >
> > > We sincerely appreciate your recognition and are especially grateful for maintaining a positive rating.
> > >
> > > Thank you for taking the time to review our manuscript and responses, as well as for your insightful comments that will greatly help us improve our work.

---

### Official Review · Reviewer_Hho8 · 2025-07-03

**Clarity:** 3
**Significance:** 3
**Originality:** 2
**Rating:** 4
**Confidence:** 4

**Summary:**

This paper proposes SignViP, a novel framework for Sign Language Video Generation (SLVG) from spoken language input.
SignViP introduces a multi-condition tokenization paradigm, leveraging both fine-grained body poses and 3D hand information to enhance generation fidelity.
Spoken language input is translated into multi-condition tokens, decoded into motion embeddings, and used to guide the video generation.

**Questions:**

1. The model heavily relies on the accuracy of the discrete multi-condition tokens generated from spoken language input, it is essential to evaluate how well this component performs. Have you measured the accuracy of these tokens?
2. FSQ is used as a core quantization method, but VQ-VAE is also commonly used in generative tokenization tasks. Could you justify the use of FSQ over more established alternatives like VQ-VAE beyond discussion in 3.1?
3. While your paper includes popular baselines like SignGAN and AnimateAnyone, would you please include more SLG works?

Q1 and Q3 addressing Significance & Quality
Q2 for Quality

**Ethical Concerns:**

["NO or VERY MINOR ethics concerns only"]

**Final Justification:**

Thanks for the response.
I have raised my score as most of my concerns have been reasonably addressed.
I recognize the contribution of this work in proposing a new representation strategy for sign language, and I acknowledge that the multi-condition setting presents a meaningful alternative to the traditional gloss space. This approach is particularly promising within the proposed multimodal pipeline.
My initial concerns primarily revolved around this design choice and its implications. While the authors have responded, I found their rebuttal somewhat underdeveloped in articulating the significance and impact of the multi-condition formulation.
In my view, the authors may have underappreciated its potential, and a more thorough discussion would have strengthened the paper.
Nonetheless, I respect differing perspectives on emphasis and framing.

My remaining concern is that the end-to-end sign language video generation significantly underperforms compared to staged approaches (such as text-to-pose and pose-to-video), and that the overall task itself appears to lag behind related work in adjacent fields, thereby limiting its potential contribution to the broader community.
Specifically, the methods generate text-to-pose have much higher back-translation scores than the methods combining end-to-end text-to-pose(including the authors' work: multi-conditions)-to-video. This observation also raises a broader question for the sign language generation domain. Across the entire SL-generation track, many subtracks, such as text-to-pose and pose-to-video, appear capable of achieving relatively strong generation performance. Why, then, does the end-to-end sign language video generation, which follows a similar conceptual pipeline (in the authors' paper) but integrates the stages into a single model, exhibit such a substantial drop in performance?

In this case, I understand that it is improper to ask authors to be responsible for field-related issues, but it is necessary to at least prove that their very low back-translation setup can reliably reflect model performance or meaningfully support model comparison.
In response, the authors further provide tables demonstrating the linear correlation between the error rate in a test set and the result of their evaluation metrics on the test set. However, this can only demonstrate effectiveness under the same distribution and does not ensure that the evaluation metric remains comparable across different distributions. From a deep learning perspective, given such low absolute scores, some parameter adjustments could lead to large fluctuations in the evaluation metrics. This makes it necessary to further examine the absence of multiple runs (of their experiments) and statistical measures such as variance in the authors’ reporting.

Overall, I believe the work is technically solid and introduces novel elements that merit recognition. However, the overall task track appears to lag behind related work in adjacent SL fields (such as text-to-pose), which suggests there may be a limit to the extent to which it contributes to the broader community. Due to this potentially constrained impact, I rate it as Borderline accept.

**Limitations:**

Yes

**Paper Formatting Concerns:**

Have not noticed.

**Quality:**

3

**Strengths And Weaknesses:**

Strengths:
1. Clarity: Clear writing and figure
2. Quality: Strong results and diverse evaluation

Weaknesses:
1. Originality & Significance: LImited techinal novelty. Its core innovation is more like combining existing concepts (tokenization, diffusion, multi condition control) in a reasonable way in SLVG tasks.
2. Quality: Some baseline are not for sign language
3. Quality: Lack of direct evaluation on token translation accuracy

---

> ### Author Rebuttal · Authors · 2025-07-31
>
> We sincerely appreciate your recognition of the paper's clear writing, as well as the quality and diversity of its evaluation results. We will address your concerns below.
>
> # Weaknesses
>
> **`W1:`** This paper lacks novelty, whose core innovation is more like combining existing concepts (tokenization, diffusion, multi condition control) in a reasonable way in SLVG tasks.
>
> **`R1:`** We respectfully disagree with your opinion and would like to further clarify the novelty and unique contributions of our paper. Our primary innovation is **a discrete tokenization paradigm through the construction of a discrete multi-condition token space** to bridge fine-grained conditions with the sign language video frames. While prior works have explored the use of multiple [1] or fine-grained [2, 3, 4] conditions for diffusion-based human video generation, there are significant distinctions compared to our method:
>
> 1. In the SLVG task, conditions are limited to serving as **intermediary** that bridge spoken language text and sign language videos. While incorporating multiple fine-grained conditions can enhance the generative process, it also increases the complexity of the translation process. To address this challenge, we introduce a compression strategy (i.e., FSQ Autoencoder) for conditions, enabling an efficient yet expressive conditional space. This strategy facilitates a balanced trade-off between translation and generation.
> 2. As a compression strategy, although quantization-based tokenization has been widely explored [5, 6, 7, 8], existing methods focus primarily on tokenizing **pixel space**. To the best of our knowledge, our work is the first to introduce tokenization on multiple fine-grained conditions for human video generation.
>
> Therefore, we believe that our core contribution is indeed novel and advances the state-of-the-art in SLVG.
>
> ---
>
> **`W2:`** Some baselines are not for sign language.
>
> **`R2:`** The baselines in Tables 1 and 2 correspond to Sign Language Video Generation (SLVG) methods and Sign Language Production (SLP) methods, respectively. In Table 3, "ControlNet" and "AnimateAnyone" are trained on sign language video datasets to evaluate their condition-to-video generative capabilities.
>
> However, we acknowledge that our paper lacks a comparison of video quality among SLVG methods. To further address the gap, we introduce two novel metrics in addition to FID and FVD, which are CLIP-FID and Identity Similarity (IDS), since temporal alignment cannot be fully preserved.
>
> Specifically, CLIP-FID is a variant of FID that leverages CLIP embeddings as frame representations. IDS assesses identity consistency between generated and ground-truth videos by calculating the cosine similarity of face embeddings extracted using YOLO5Face and Arc2Face.
>
> The following experimental results validate the state-of-the-art performance of SignViP.
>
> | *[RWTH-2014T Dataset]* | FID (↓) | CLIP-FID (↓) | FVD (↓) | IDS |
> | :- | :-: | :-: | :-: | :-: |
> | SignGAN | 547.90 | 167.70 | 1431.38 | 0.463 |
> | SignGAN w/ AnimateAnyone | 595.99 | 161.97 | 1330.54 | 0.462  |
> | SignGen | 644.06 | 184.66 | 1715.32 | 0.515 |
> | **SignViP (Ours)** | **508.91** | **154.10** | **1025.45** | **0.571** |
>
> | *[How2Sign Dataset]* | FID (↓) | CLIP-FID (↓) | FVD (↓) | IDS |
> | :- | :-: | :-: | :-: | :-: |
> | SignGAN | 667.44  | 210.11 | 2766.97 | 0.538 |
> | SignGAN w/ AnimateAnyone | 679.41 | 215.05 | 2484.39 | 0.533  |
> | SignGen | 815.69  | 186.32 | 3538.49 | 0.539 |
> | **SignViP (Ours)** | **575.67** | **109.61** | **2207.67** | **0.624** |
>
> ---
>
> **`W3:`** Lack of direct evaluation of token translation accuracy.
>
> **`R3:`** It is challenging to directly compute the accuracy of token translation. Because of the temporal shifts and uneven scaling, the translated tokens during inference may not perfectly align with ground-truth tokens. As a replacement, we employ **normalized DTW distance** as the metric.
>
> **Dynamic Time Warping (DTW)** distance is a similarity measure that computes the optimal alignment between two sequences, even if they differ in length or are not aligned one-to-one. The DTW distance is normalized to accommodate variations in sequence length.
>
> For our method, we decode the translated tokens into condition embeddings using the decoder of the FSQ Autoencoder, and compare these embeddings with the ground-truth condition embeddings using the normalized DTW distance.
>
> To validate the effectiveness of normalized DTW distance, we follow the experimental settings in **Section 4.3 (Effect of Compression)** and **Appendix D.7 (Effect of Scheduled Sampling Strategy)**. The following experimental results show that the trend of the normalized DTW distance closely matches that of previously reported metrics, confirming its reliability as an evaluation metric for token translation.
>
> | Compression Rate | Norm. DTW Distance (↓) | BLEU-4 | ROUGE |
> | :- | :-: | :-: | :-: |
> | 1 | 1.9020 | 1.32 | 17.38 |
> | 2 | 1.7937 | 4.14 | 21.89 |
> | 4 | 1.7438 | 5.64 | 23.68 |
> | **8 (Ours)** | **1.5968** | **8.65**| **28.85** |
> | 16 | 1.7291 | 7.02 | 24.72 |
>
> | Sampling Rate | Norm. DTW Distance (↓) | BLEU-4 | ROUGE |
> | :- | :-: | :-: | :-: |
> | 0.2 | 1.6123 | 8.12 | 27.72 |
> | **0.4 (Ours)** | **1.5968** | **8.65** | **28.85** |
> | 0.6 | 1.5978 | 8.30 | 28.23 |
> | 0.8 | 1.6191 | 7.82 | 27.15 |
> | 1.0 | 1.7354 | 5.89 | 24.71 |
>
>
> # Questions
>
> **`Q1:`** The model relies on the accuracy of the discrete multi-condition tokens. Have you measured the accuracy of these tokens?
>
> **`A1:`** Please refer to our response to Weakness 3.
>
> ---
>
> **`Q2:`** Could you justify the use of FSQ over more established alternatives like VQ-VAE?
>
> **`A2:`** In **Appendix D.8 (FSQ vs. VQ)**, we have compared FSQ and VQ as quantization strategies for the FSQ Autoencoder. Specifically, we evaluated both methods in terms of **codebook usage** efficiency and multi-conditional **reconstruction loss**.
>
> The results are summarized as follows: the codebook usage experiments demonstrate the stability and scalability of FSQ, while the reconstruction loss experiments show that FSQ more effectively preserves the original conditional structure.
>
> | Codebook Size | Usage% (FSQ) | Usage% (VQ) | Recon. Loss (FSQ) (↓) | Recon. Loss (VQ) (↓) |
> | :- | :-: | :-: | :-: | :-: |
> | $2^7$ | 100% | 90.6% | 0.1545 | 0.1569 |
> | $2^8$ | 100% | 51.6% | 0.1389 | 0.1545 |
> | $2^9$ | 99.4% | 26.6% | 0.1268 | 0.1528 |
> | $2^{10}$ | 99.9% | 14.6% | 0.1270 | 0.1512 |
> | $2^{11}$ | 99.2% | 6.9% | 0.1188 | 0.1520 |
> | $2^{12}$ | 97.81% | 3.6% | 0.1160 | 0.1511 |
>
> Additionally, in **Section 4.3 (Effect of Quantization)**, we abalated quantization from FSQ Autoencoder, treating this as a special quantization strategy. The qualitative comparison in Figure 7 demonstrates that ablating quantization leads to weak semantic alignment and low video quality. It highlights the importance of quantization for our SignViP. Here, we present the detailed quantitative results on RWTH-2014T.
>
> | *[Video Back-Translation]* | BLEU-1 | BLEU-4 | ROUGE |
> | :- | :-: | :-: |:-:|
> | w/o FSQ | 9.43 | 0.42 | 13.58 |
> | **w/ FSQ (Ours)** | **26.72** | **8.65** | **28.85** |
>
> | *[Pose Back-Translation]* | BLEU-1 | BLEU-4 | ROUGE |
> | :- | :-: | :-: | :-: |
> | w/o FSQ | 8.11 | 0.30 | 11.29 |
> | **w/ FSQ (Ours)** | **21.94** | **4.61** | **22.67** |
>
> | *[Video Quality]* | FVD (↓) | SSIM   | PSNR  | LPIPS (↓) | Hand SSIM |
> | :- | :-: | :-: | :-: | :-: | :-: |
> | w/o FSQ | 1932.26 | 0.473 | 10.02 | 0.434 | 0.258 |
> | **w/ FSQ (Ours)** | **1025.45** | **0.609** | **13.43** | **0.331** | **0.294** |
>
> ---
>
> **`Q3:`** Would you please include more SLVG works as baselines?
>
> **`A3:`** Thank you for your suggestion.
>
> We add new baselines that combine SLP methods (i.e., MDN and MoMP) with pose-to-video methods (i.e., ContorlNet and AnimateAnyone). Specifically, we first use an SLP method to convert spoken language text into skeleton poses, and then employ a pose-to-video method to synthesize sign language videos from these poses.
>
> The video back-translation results are presented below, which demonstrate that our SignViP still outperforms the newly added baselines.
>
> | *[RWTH-2014T Dataset]* | BLEU-1 | BLEU-4 | ROUGE |
> | - | :-: | :-: | :-: |
> | MDN+ControlNet | 18.52 | 3.25 | 20.88 |
> | MDN+AnimateAnyone | 19.45 | 3.33 | 21.02 |
> | MoMP+ControlNet | 19.12 | 3.61 | 21.54 |
> | MoMP+AnimateAnyone | 20.05 | 3.72 | 21.68 |
> | **SignViP (Ours)** | **26.72** | **8.65** | **28.85** |
>
>
> | *[How2Sign Dataset]* | BLEU-1 | BLEU-4 | ROUGE |
> | - | :-: | :-: | :-: |
> | MDN+ControlNet | 11.95 | 2.09 | 13.05 |
> | MDN+AnimateAnyone | 13.08 | 2.03 | 13.48 |
> | MoMP+ControlNet | 12.53 | 2.31 | 13.72 |
> | MoMP+AnimateAnyone | 13.65 | 2.25 | 14.15 |
> | **SignViP (Ours)** | **16.21** | **5.04** | **16.99** |
>
> # References
>
> `[1]` Follow-Your-Pose V2: Multiple-Condition Guided Character Image Animation for Stable Pose Control, arXiv:2406.03035.
> `[2]` Magicanimate: Temporally Consistent Human Image Animation Using Diffusion Model, CVPR 2024.
> `[3]` Magicportrait: Temporally Consistent Face Reenactment with 3D Geometric Guidance, arXiv:2504.21497.
> `[4]` Vlogger: Multimodal Diffusion for Embodied Avatar Synthesis, arXiv:2403.08764.
> `[5]` Finite Scalar Quantization: VQ-VAE Made Simple, ICLR 2024.
> `[6]` Language Model Beats Diffusion–Tokenizer is Key to Visual Generation, arXiv:2310.05737.
> `[7]` Image and Video Tokenization with Binary Spherical Quantization, arXiv:2406.07548.
> `[8]` Infinity: Scaling Bitwise AutoRegressive Modeling for High-Resolution Image Synthesis, arXiv:2412.04431.

---

### Official Review · Reviewer_g4KT · 2025-07-05

**Clarity:** 2
**Significance:** 2
**Originality:** 3
**Rating:** 4
**Confidence:** 3

**Summary:**

The paper introduces SignViP, a new framework for Sign Language Video Generation from spoken language texts. Unlike previous approaches that rely on a single coarse condition, SignViP leverages multiple fine-grained conditions, such as fine-grained body poses and 3D hands, to improve the naturalness and expressiveness of generated sign language videos. Experimental results on two different datasets demonstrate that SignViP achieves state-of-the-art performance in video quality, temporal coherence, and semantic fidelity.

**Questions:**

Is there any other evaluation metric to determine the quality of video generation? As BLEU and Rouge scores have already shown not perfect in language generation tasks.

**Ethical Concerns:**

["NO or VERY MINOR ethics concerns only"]

**Final Justification:**

Most of my concerns have been addressed. The additional experimental results further verifiy the effectiveness of the proposed method. I have raised the score accordingly.

**Limitations:**

yes

**Quality:**

3

**Strengths And Weaknesses:**

Strengths:
1. Clear motivation to incorporate multiple fine-grained conditions.
2. Achieving new SOTA results on two evaluation datasets when using video back-translation.

Weaknesses:
1. The method introduction is a bit complicated, without emphasizing the novel part of the paper.
2. The performance evaluation mainly depends on back-translation techniques, which highly relates to the quality of back-translation model. While the method achieves SOTA on Table 1, several metrics in Table 2 fall behind MoMP method.
3. Sign Video Diffusion Model is initialized with an existing diffusion model, and it lacks ablation to show the effect of such initialization, making whether the comparison is fair is not sure.

---

> ### Author Rebuttal · Authors · 2025-07-31
>
> We sincerely appreciate your recognition of the clear motivation and SOTA performance demonstrated in our work. We will address your concerns below.
>
> # Weaknesses
>
> **`W1:`** The method introduction is a bit complicated, without emphasizing the novel part of the paper.
>
> **`R1:`** The main novelties of our paper are detailed in Sections 3.2 and 3.3. Our key innovations are as follows:
>
> 1. **Discrete Tokenization Paradigm** (lines 144–145): To mitigate errors in directly translating high-dimensional conditions, we propose a discrete tokenization approach to integrate and represent these conditions;
> 2. **Novel Module Design** (lines 146–154): We present new modules specifically designed for SignViP, enabling more robust processing of multi-condition inputs;
> 3. **Multi-Stage Training Strategy** (Section 3.3): We introduce a multi-stage training strategy to construct a comprehensive multi-condition token space for our discrete tokenization paradigm.
>
> We will improve the method section to more clearly emphasize these core innovations and improve its clarity and readability.
>
> ---
>
> **`W2:`** The performance evaluation mainly depends on back-translation techniques, which highly relate to the quality of the back-translation model.
>
> **`R2:`** We agree that the evaluation protocol is highly dependent on the quality of the back-translation model, which may introduce bias. While human evaluation maybe provide a more reliable resolution, it is challenging to recruit sign language experts.
>
> To address this, we further adopt a compromise human evaluation strategy that does not require the sign language experts. Specifically, we present human evaluators with the ground-truth video alongside several corresponding anonymized videos generated by different SLVG methods. The evaluators are given the following instruction:
>
> "*Please choose the video where the signer's actions appear most similar to those in the ground-truth video.*"
>
> We invited 10 evaluators, each of whom received 25 group samples for the RWTH-2014T dataset and 50 for the How2Sign dataset, resulting in totals of 250 and 500 votes, respectively. We calculate the proportion of votes for each SLVG method, with the results shown below. Notably, our SignViP receives the highest proportion of votes, indicating that SignViP generates more accurate sign language videos.
>
> | | #Votes (RWTH) | Vote% (RWTH) | #Votes (H2S) | Vote% (H2S) |
> | :- | :-: | :-: | :-: | :-: |
> | SignGAN | 25 | 10.0% | 31 | 6.2% |
> | SignGAN w/ AnimateAnyone | 23 | 9.2% | 24 | 4.8% |
> | SignGen | 27 | 10.8% | 36 | 7.2% |
> | **SignViP (Ours)** | **175** | **70.0%** | **409** | **81.8%** |
> | *TOTAL* | *250* | *100.0%* | *500* | *100%* |
>
> ---
>
> **`W3:`** While the method achieves SOTA on Table 1, several metrics in Table 2 fall behind the MoMP method.
>
> **`R3:`** In Table 2, we compare our SLVG  method with SLP (Sign Language Production) approaches rather than other SLVG methods. These SLP baselines translate text directly into pose sequences, which aligns with our pose back-translation evaluation pipeline. However, for our SLVG method, we need to detect poses from the generated videos, which may introduce additional errors and impact the evaluation results. Consequently, the state-of-the-art SLP method (i.e., MoMP) outperforms our method on certain metrics in Table 2, although our results remain comparable.
>
> To enable a fair comparison in the SLVG setting, we further combine MoMP with a pose-to-video generation approach. The following video back-translation results demonstrate that our method is better suited for the SLVG task compared to MoMP-based methods.
>
> | *[RWTH-2014T Dataset]* | BLEU-1 | BLEU-2 | BLEU-3 | BLEU-4 | ROUGE |
> | :- | :-: |:-: | :-: | :-: | :-: |
> | MoMP+ControlNet | 19.12 | 8.95 | 5.33 | 3.61 | 21.54 |
> | MoMP+AnimateAnyone | 20.05 | 8.79 | 5.24 | 3.72 | 21.68 |
> | **SignViP (Ours)** | **26.72** | **15.65** | **11.14** | **8.65** | **28.85** |
>
> | *[How2Sign Dataset]* | BLEU-1 | BLEU-2 | BLEU-3 | BLEU-4 | ROUGE |
> | :- | :-: | :-: | :-: | :-: |:-: |
> | MoMP+ControlNet | 12.53 | 5.59 | 3.48 | 2.31 | 13.72 |
> | MoMP+AnimateAnyone | 13.65 | 5.82 | 3.39 | 2.25 | 14.15 |
> | **SignViP (Ours)** | **16.21** | **9.36** | **6.28** | **5.04** | **16.99** |
>
> ---
>
> **`W4:`** Sign Video Diffusion Model is initialized with an existing diffusion model, and it lacks ablation to show the effect of such initialization, making whether the comparison is fair is not sure.
>
> **`R4:`** The comparison is fair, because all the diffusion-based SLVG methods (i.e., SignGAN+AnimateAnyone, SignGen, and our SignViP) are initialized with parameters from existing pretrained diffusion models.
>
> Training a video generation diffusion model from scratch is extremely challenging due to the high computational costs and slow convergence. Therefore, using pretrained diffusion parameters to initialize the customized diffusion model is a fundamental strategy to significantly accelerate training [1, 2, 3, 4, 5].
>
> To further validate the effectiveness of pretrained parameter initialization, we conducted an ablation study, evaluating the quality of generated videos both with and without pretrained initialization under the same training steps. The results below clearly demonstrate the substantial advantage of initializing with pretrained parameters.
>
> | | FID (↓) | FVD (↓) |
> | :- | :-: | :-: |
> | w/o Pretrained Initialization | 2278.23 | 3277.20 |
> | **w/ Pretrained Initialization (Ours)** | **508.91** | **1025.45** |
>
> # Questions
>
> **`Q1:`** Is there any other evaluation metric to determine the quality of video generation? As BLEU and ROUGE scores have already shown not perfect in language generation tasks.
>
> **`A1:`** Thank you for your suggestion. To provide a more comprehensive evaluation of back-translated texts, we further employ the COMET [6, 7] metric, which is specifically designed to predict human judgments of machine translation quality. COMET is widely used for machine translation tasks [8, 9, 10] and is considered more suitable than BLEU and ROUGE.
>
> As shown below, for video back-translation, our SignViP still outperforms other SLVG baselines according to the COMET metric.
>
> | | COMET (RWTH) | COMET (H2S) |
> | :- | :-: | :-: |
> | *Ground-Truth* | *0.6157* | *0.5882* |
> | SignGAN | 0.4977 | 0.5104 |
> | SignGAN w/ AnimateAnyone | 0.4928 | 0.5135 |
> | SignGen | 0.4086 | 0.4127 |
> | *MoMP+ControlNet* | 0.5033 | 0.5122 |
> | *MoMP+AnimateAnyone* | 0.5091 | 0.5208 |
> | **SignViP (Ours)** | **0.5608** | **0.5524** |
>
> As shown below, for pose back-translation, SignViP achieves performance that is slightly lower but comparable to the state-of-the-art SLP method (i.e., MoMP). However, when MoMP is combined with pose-to-video generative models (i.e., ControlNet or AnimateAnyone) for the SLVG task, the resulting MoMP-based methods yield lower COMET scores than our approach. These results on the COMET metric indicate that SignViP is more suitable for the SLVG task compared to MoMP-based methods.
>
> | | COMET (RWTH) | COMET (H2S) |
> | :- | :-: | :-:|
> | *Ground-Truth* | *0.5978* | *0.6250* |
> | ProTran | 0.5091 | 0.5549 |
> | Adversarial | 0.5127 | 0.5618 |
> | MDN | 0.5251 | 0.5685 |
> | *MoMP* | **0.5466** | **0.5802** |
> | **SignViP (Ours)** | 0.5347 | 0.5738 |
>
> # References
>
> `[1]` Latent Video Diffusion Models for High-Fidelity Long Video Generation, arXiv:2211.13221.
> `[2]` Stable Video Diffusion: Scaling Latent Video Diffusion Models to Large Datasets, arXiv:2311.15127.
> `[3]` Customize-A-Video: One-Shot Motion Customization of Text-to-Video Diffusion Models, ECCV 2024.
> `[4]` AnimateDiff: Animate Your Personalized Text-to-Image Diffusion Models without Specific Tuning, ICML 2024.
> `[5]` Follow Your Pose: Pose-Guided Text-to-Video Generation Using Pose-Free Videos, AAAI 2024.
> `[6]` COMET-22: Unbabel-IST 2022 Submission for the Metrics Shared Task, WMT 2022.
> `[7]` COMET: A Neural Framework for MT Evaluation, EMNLP 2020.
> `[8]` Hallucinations in Large Multilingual Translation Models, TACL 2023.
> `[9]` Tower: An Open Multilingual Large Language Model for Translation-Related Tasks, COLM 2024.
> `[10]` Exploring Human-Like Translation Strategy with Large Language Models, TACL 2024.

---

> > ### Author Response · Authors · 2025-08-04
> >
> > Dear Reviewer g4KT,
> >
> > We have carefully prepared a response to address your concerns.
> > Could you kindly take a moment to review it and let us know if it resolves the issues you raised?
> > If you have any additional questions or suggestions, we would be happy to address them.
> >
> > Thank you for your time and consideration.
> >
> > The Authors

---

> > ### Comment · Reviewer_g4KT · 2025-08-06
> >
> > Thank you for your response. Most of my conserns have been addressed. I will raise the score accordingly.

---

> > > ### Author Response · Authors · 2025-08-07
> > >
> > > Thank you for taking the time to review our manuscript and our responses, as well as for your insightful feedback that has greatly helped us improve our work.
> > >
> > > We sincerely appreciate your recognition and are especially grateful for the increased score.

---

### Decision · Program_Chairs · 2025-09-17

**Decision:**

Accept (spotlight)

**Comment:**

The submission proposed a method on text to Sign Language Video. To generate the video details in multiple scale and key components, multiple fine-grained conditions are applied. It provides details from finger, face to body pose. However, the generated video is not smooth enough from frame to frame. A stabilizer will be helpful to generate the video. Before rebuttal, this submission received two borderline rejects and two accepts. Both reviewers and authors engaged in discussion, the rebuttal persuaded the reviewers and the two reviewers upgraded their rating to borderline accepts. Most of the concerns are well addressed. The AC recommends accepting this submission.